# Formulation of Compressed Earth Blocks Stabilized by Glass Waste Activated with NaOH Solution

**Sihem Larbi** [1,*] , **Abdelkrim Khaldi** [1] , **Walid Maherzi** [2,*] **and Nor-Edine Abriak** [2]

1 Laboratory of Rheology, Transport and Treatment of Complex Fluids (LRTTFC), Hydraulics Department, University of Sciences and Technology of Oran Mohamed Boudiaf (USTO-MB), Oran 31000, Algeria; khaldiakz@yahoo.fr
2 Laboratory of Civil Engineering and Geo-Environment LGCgE, Materials and Process Department, IMT Nord Europe, F-59000 Lille, France; nor-edine.abriak@imt-nord-europe.fr
* Correspondence: sihem.larbi@univ-usto.dz (S.L.); walid.maherzi@imt-nord-europe.fr (W.M.)

**Abstract:** Due to the increase in demand for building materials and their high prices in most developing countries, many researchers are trying to recycle waste for use as secondary raw materials. The aim of this study is the optimization of a mixture of compressed earth blocks based on two sediments. These sediments were tested through the Vicat test to determine the proportion of each one and the optimal water content. The mixtures were treated by adding 10% of blast furnace slag and different proportions of dissolved glass in a NaOH solution. The results indicated that the mixture of 70% Oran sediments with 30% Sidi Lakhdar sediments treated with 4% glass waste produced a CEB (compressed earth block) with high compressive strength with low porosity. In addition, formulated CEBs have a very good resistance to water immersion.

**Keywords:** sediments; glass; NaOH concentration; compressed earth blocks; circular economy





## 1. Introduction

Low-cost construction is a modern civil engineering concept that uses locally available materials to obtain the desired strength, performance, and durability [1]. The current construction rate in developing countries is generally insufficient to meet the needs of only a 10% net population increase per year [2]. Therefore, it is necessary to introduce new modern techniques in building construction by using new materials such as compressed earth blocks (CEB), which are a form of a ground construction unit based on the use of local materials [3], stabilized and pressurized to form a soil block [3]. This is an available material that is recyclable as raw materials in cases of improper treatment [4]. The energy needed for their manufacture is also very low. When local sectors are available, the impact associated with transportation is negligible. Moisture is the main barrier to using ground bricks [5]. The soil is mixed with an adjuvant, and sometimes Portland cement [6] or hydrated lime is added at a consistent ratio to increase weather resistance [7]. The consumption of CEB in Algeria has increased by around 7.3 million m$^3$/year [8]. Sediments raise many problems through the concentration of pollution and the movement of potentially dangerous pollutants [9]. Dredging deposits are considered waste [10] rather than raw materials [11]. Dredged sediments are one of the largest potential waste streams in Algeria, with an annual production of about 10 million m$^3$ [12]. Inland water bodies in Algeria are estimated to produce 1.9 billion m$^3$: 375 Hm$^3$/year in Oran and 27 Hm$^3$/year in Mostaganem [13]. Recently, several studies have explored the possibility of reusing dredging sediments as alternative materials for different applications: such as a mineral addition to cement [14], as lightweight aggregates [10], raw materials for road construction [15], or as cementation materials [11], geopolymers [16], and bricks and tiles [16,17]. One innovative solution to recovering dredged sediments is the use of environmental binders, also called geopolymer binders or alkaline active binders. Miranda et al. [18] used

two alkaline-activated floor mortars, and their results showed that a significant mechanical strength improvement was obtained in terms of the compression and shear behavior of the formulated materials. According to Narayanaswamy et al. [19], alkali-activated compacted earth blocks appear to be very promising for reducing the global warming potential for the construction sector. Omar Sore et al. [20] evaluated the feasibility of CEB stabilization with a geopolymer made from a mixture of metakaolin solution and sodium hydroxide. Their results demonstrate that geopolymer binders significantly improved the mechanical performance of CEB and endowed it with thermal properties that were almost the same as those of the unstable blocks. Bouchikhi et al. [21] showed that glass waste was an active solution with a good geopolymer binder-reaction production ability that enhances the properties of other materials. Our study investigated the production of CEB, comprised of a glass powder additive and NaOH solution [22]. This alternative use would divert the material away from landfills [23]. The use of glass powder in CEB production can make the construction industry more sustainable. Glass powder has pozzolanic characteristics [24]. This solution gives great chemical stability [25] due to the covalent or ionic bonds which unite the atoms, as well as good biocompatibility [26].

## 2. Materials and Methods

### 2.1. Materials

In this work, sediments dredged from the two Algerian ports of Oran and Sidi Lakhdar were used for the formulation of CEBs (Figure 1). Oran is the largest port on the southern Mediterranean shore housing both fishing and commercial activities [17], in which the optimal deep docking depth to accommodate large cargo ships has been reduced by one to two meters in depth [27] due to sediment deposits occurring at a rate of 69.704 $m^3$/day [28,29]. This represents a real obstacle to the development of economic activity and may require the dredging and treatment of 120.000 $m^3$ of dredged sediment. Moreover, the uncontrolled discharge from urban and industrial activities, estimated at 1325 L/s, [30] causes sediment pollution. The Sidi Lakhdar port, located in the eastern region of Mostaganem, mainly polluted by wastewater discharge into Wadi Obeid [31], has sediment deposit problems estimated at 280.000 $m^3$ of sediments needing to be dredged, which currently make the docking of fishing boats and vessels increasingly difficult [32].

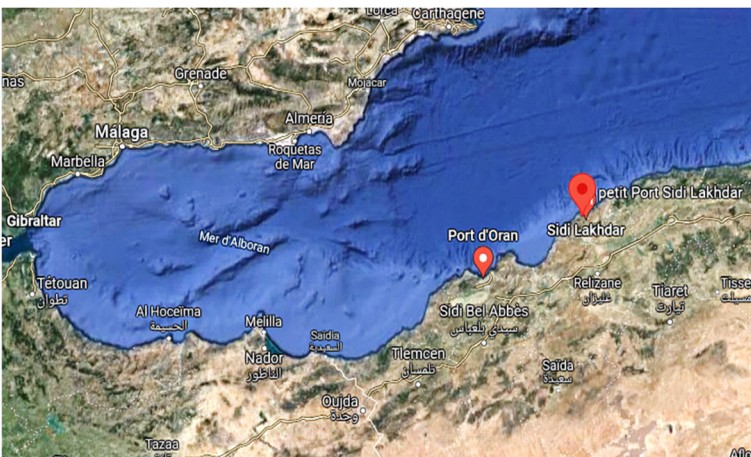

**Figure 1.** Representation of the study area.

2.1.1. Physicochemical Characterization

The samples were analyzed through several physicochemical and mineralogical tests (Table 1), to classify and compare the characteristics and behaviors of the sediments, glass powder and blast furnace slag (BFS).

**Table 1.** Physicochemical characterization of materials.

| Characterization | | Sediments | | Glass Powder | BFS |
|---|---|---|---|---|---|
| | | Oran | Sidi Lakhdar | | |
| Water content (%) | | 6.35 | 0.36 | 39 | 33 |
| Density Gs (kg/m$^3$) | | 2500 | 2670 | 2540 | 2910 |
| d10 (μm) | | 0.7 | 10 | 10 | 0.1 |
| d50 (μm) | | 3.5 | 15 | 50 | 1 |
| d90 (μm) | | 100 | 40 | 160 | 3 |
| Organic matter (%) | | 7.25 | 0.92 | | |
| LOI (%) | | 15.65 | 6.72 | 0.03 | |
| Methylene blue value (%) | | 0.62 | 0.5 | | |
| Atterberg limit | Wp (%) | 20.97 | | | |
| | Wl (%) | 35.4 | 24.33 | | |
| | Ip (%) | 14.4 | | | |
| Sand equivalent (%) | | | 49.9 | | |
| Specific surface (m$^2$/kg) | | 14,000 | 10,500 | 792.6 | 493 |
| Chemical components (%) | Fe$_2$O$_3$ | 3.58 | 3.5 | 0.41 | 0.5 |
| | Al$_2$O$_3$ | 1.6 | 4.3 | 1.61 | 10.8 |
| | TiO$_2$ | 0.2 | 0.2 | Traces | 0.7 |
| | SiO$_2$ | 15.2 | 43 | 70.86 | 38 |
| | CaO | 28.7 | 10.7 | 11.52 | 42.5 |
| | P$_2$O$_5$ | 0.3 | 0.13 | Traces | |
| | MgO | 2.64 | 1.2 | 1.18 | 6.6 |
| | K$_2$O | 0.04 | 0.6 | 0.69 | 0.35 |
| | CaCO$_3$ | 39.71 | 14.01 | | |
| | Na$_2$O | | | 13.58 | 0.28 |
| | Cr$_2$O$_3$ | | | 0.15 | |
| Heavy metals (%) | As | <0.06 | <0.06 | | |
| | Ba | 0.26 | 0.039 | | |
| | Cd | <0.007 | <0.007 | | |
| | Cr | <0.005 | 0.005 | | |
| | Cu | 0.19 | 0.012 | | |
| | Mo | 0.078 | <0.06 | | |
| | Ni | <0.05 | <0.05 | | |
| | Pb | <0.05 | <0.05 | | |
| | Sb | <0.06 | <0.06 | | |
| | Se | <0.09 | <0.09 | | |
| | Zn | <0.04 | <0.04 | | |
| | Fluorides | 42 | 17 | | |
| | Chlorides | 9760 | 218 | | |
| | Sulfates | 3580 | 174 | | |
| pH | | 6.6 | 6.8 | | |
| Conductivity (mS/cm) | | 10.21 | 0.36 | | |

The Casagrande method describes the plastic behavior of the samples by determining the Atterberg limits [33]. The liquidity limit (Ll), the plasticity limit (LP) and therefore the plasticity index (PI) were found to be consistent with the standard [34,35]. Before analysis, the raw materials were dried inside a laboratory oven at 105 °C for 24 h [36,37]. In order to measure the liquidity limit (L1) and hence the plasticity limit (LP), the samples were sieved at 400 μm [38,39]. The precise gravity of solid grains Gs was determined using an ACCUPYC 1330 helium gas pycnometer consistent with European standard NF EN ISO 8130-2 [40,41]. The chemical composition of the sediments was determined using a BRUCKER S4 for X-ray fluorescence spectrometry measurements. The sample was burned

at 550 °C for three hours, oxidizing the organic matter and transforming it into carbon dioxide ($CO_2$) and water vapor. After combustion, only the mineral fraction of the soil remained in the container, in accordance with the NF EN 15,935 standard [42,43]. The method for determining the organic matter consists of analyzing the raw sediments, and therefore the preparation of the sediment fraction after centrifugation was oriented at random: dried in air (at room temperature), saturated with glycol (EG), and heated at 550 °C for 1 h. Increasing the organic matter content increases plasticity and secondary compression and reduces permeability. According to Varghese et al. [44], the effect of the organic matter content outweighs that of the geotechnical behavior when it exceeds 4 to 5%.

The precise surface was obtained by the following equation, which was consistent with that used in previous works like Laribi et al. [45]:

$$S = ((VB/100) \times (N/373)) \times 130 \times 10^{-20} \tag{1}$$

where $N$ is number of Avogadro = $6.023 \times 10^{23}$ and $VB$ is the weight of blue in the liquid $\times$ 100 g (reduced to 100 g of material) $\times$ 0.010 (dilution of blue)/weight of the dry sample.

The morphological characteristics of the sediments were analyzed by a Hitachi S-3600 N scanning microscope. The thermogravimetric analysis (TGA) was carried out by a NESTZSCH STA 449F3 instrument and the mass loss depending on the temperature measurements between 105 °C and 1000 °C was registered [46]. The chemical analysis of the sediments shows that the percentage of silica is very high and calcium was relatively high, so this material was rich in Calcite ($CaCO_3$). The alumina/silica ratio provides information on the permeability of the material. The greater this ratio, the greater the permeability [47]. In our case, this ratio was small: $Al_2O_3/SiO_2 = 0.1$. The $SiO_2/Al_2O_3$ molar ratio was greater than the conventional value for bentonite, which was 2.5. This difference indicates the presence of free quartz inside the clay fraction in huge percentage [34,36]. The overall composition of the other oxides ($Fe_2O_3$, MgO, $K_2O$ and $Na_2O$) reached a percentage of 2.68 in the sediments from the port of Oran and 5.3 in sediments from the port of Sidi Lakhdar, which shows that the sediments were not pure [48]. Sediments with a relatively low molar ratio ($CaO/SiO_2$) have a relatively high electrical conductivity. In addition, their flexural strength gradually increases with a decrease in the $CaO/SiO_2$ molar ratio [49]. Loss of ignition (LOI) is the result of the calcination of powders up to 1000 °C. The high LOI was related to the presence of carbonates and the water evaporated by heat treatment. This result explains that the water retention due to the large capillarity of the large surface area. Texture analysis (Figure 2) to determine the particle size distribution was conducted by the wet sieving method [50].

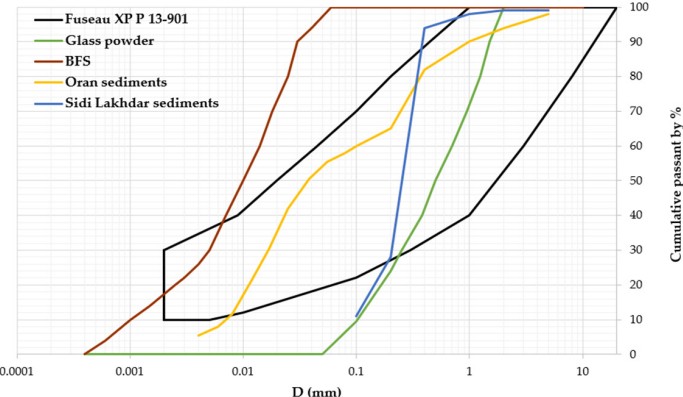

**Figure 2.** Particle size distribution of materials.

The particle size analysis of the two sediments entered the zone of the soil texture diagram according to the standard XP P13-901 [51] which means that the two sediments gave satisfactory results.

### 2.1.2. Mineralogical Identification

The mineralogical structure of varied sediments has been studied. The centrifugation technique makes the separation of minor fractions from each sample possible.

- XRD analysis.

The samples were placed in oriented slides, then scanned in dry air, treated with glycol solvation, and heated to 550 °C. The mineralogical composition of the samples was determined by X-ray diffraction using an energy dispersion BRUKER AXS D8 Advance, to see the mineral phases of the material [52,53].

The mineralogical composition of the sediments was that of quartz (Q) and calcite (C) impurities for both Oran and Sidi Lakhdar sediments (Figure 3).

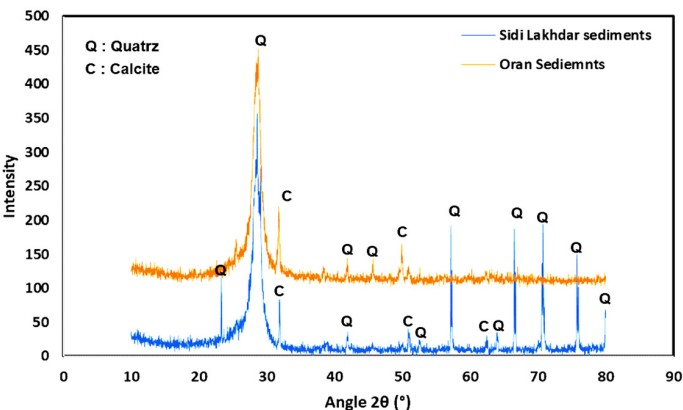

**Figure 3.** XRD mineralogical analysis of raw sediments.

- Thermogravimetric Analysis (TGA) of Raw Sediments.

Figure 4 shows the thermogravimetric (TGA) and differential thermal (DTA) analysis curves. The TGA curves reveal four successive mass losses in relation to the four peaks of differential thermal analysis observed in the intervals of successive temperatures. The first interval, from room temperature up to 180 °C, saw a mass loss of 0.52% of the total mass of the sample, with a very low amplitude peak observed on the DTA curve. The second mass loss of 1.63% began at around 180 °C and extended up to 540 °C. This was smallest mass loss, the reason for which can be attributed to the desorption of water from the material structure and the decomposition of organic matter. The DTA weak amplitude peak observed between 380 °C and 540 °C results from the superposition of a reaction due to the desorption of structural water and from other reactions that resulted from the combustion of volatile organic matter [54].

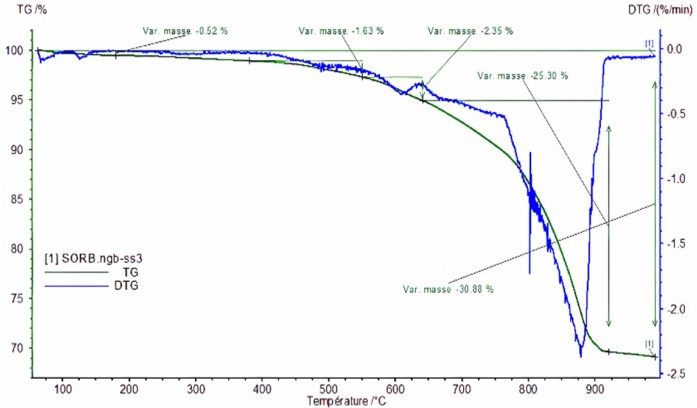

**Figure 4.** Thermogravimetric analysis of Oran sediment.

The third domain was between 540 °C and 630 °C. In this interval, the mass loss was 2.35% of the total mass of the sample. A greater mass loss of 25.3% was observed between 630 °C and 1000 °C, which was associated with an endothermic peak due to the decomposition of calcite ($CaCO_3$) [55]. The DTA peak observed around 880 °C was attributed to this loss of mass due to the dehydroxylation of clay minerals. At 920 °C, the overall loss was around 29.8%.

The TGA and DTA curves of Sidi Lakhdar sediments are shown in Figure 5. The TGA curve shows four successive mass losses in relation to four temperature ranges. The first range was from room temperature to 180 °C, at which point the sample had lost 1.05% of its total mass. This loss corresponds to the desorption of water molecules from the sediment surface. The peak observed on the DTA curve at 120 °C confirms that this loss was due to the expulsion of adsorbed water. The second domain was between 180 °C and 380 °C. The sample lost 1.24% of its mass. The peak observed on the DTA curve at 300 °C was due to the structural departure of water and the decomposition of organic matter [56].

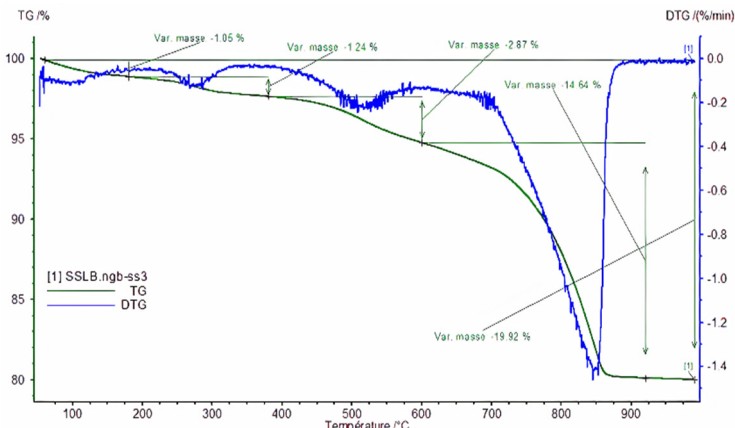

**Figure 5.** Thermogravimetric analysis of Sidi Lakhdar sediment.

The third range was between 380 °C and 600 °C. On the TGA curve, a loss of 2.87% of the total mass of the sample was detected in this interval. The peak observed at around 500 °C on the DTA curve allowed this loss to be attributed to the dehydroxylation of clay minerals [57–59].

The last loss interval between 600 °C and 1000 °C on the TGA curve corresponds to a mass loss of 14.64%. This loss is associated with a peak DTA curve around 850 °C, and was due to the decomposition of carbonates and the release of $CO_2$ [60].

2.1.3. Physical and Chemical Properties of Sodium Hydroxide

Sodium hydroxide (NaOH) (Table 2) is a corrosive white crystalline solid that readily absorbs moisture until it dissolves. Commonly called caustic soda, or lye, sodium hydroxide is the most widely used industrial alkali [61].

**Table 2.** Chemical characterization of sodium hydroxide.

| Chemical Formula | NaOH |
| --- | --- |
| Density | 2.13 |
| Molecular weight | 40.01 |
| Melting point | 318 °C |
| Boiling point | 1390 °C |
| Solubility | Soluble in water, ethanol and glycerol |

*2.2. Methods*

2.2.1. Determination of the Amount of Sediment Added

The raw brick samples (Figure 6) were obtained by mixing the two sediments at different percentages to obtain the optimal amount of the mixture.

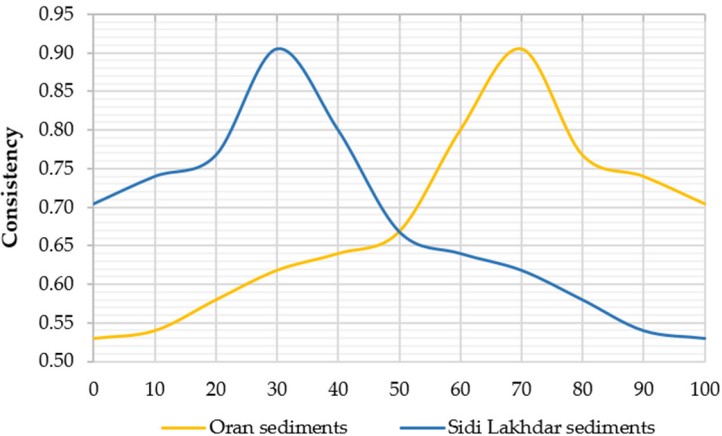

**Figure 6.** Consistency curve as a function of percentage of the Oran port sediment.

The normal consistency based on the VICAT was tested according to the EN 196-2 standard [62,63] of the different percentages of sediments, and is compactness was determined by the following relation [64,65]:

$$C = \frac{1000}{1000 + Mv\frac{Me}{Mp}} \tag{2}$$

where $Mv$ is the actual density of the powder (kg/m$^3$) of a binary mixture, $Me$ is the amount of water (kg), and $Mp$ is mass of powder (kg).

The VICAT test of the different percentages of sediments gave an optimal consistency of 0.905, indicating that the best mixture consists of 70% of sediments from the port of Oran and 30% from the sediments from the port of Sidi Lakhdar.

2.2.2. Determination of the Optimal Water Content

Water was added to the mixture of two sediments (70% sediment from the port of Oran and 30% sediment from the port of Sidi Lakhdar) until it comprised 8, 10, 13 and 19% of the mixture in order to determine the optimal water content. Pressure of 10 MPa was applied, creating a force of 1962 daN on a surface of 19.62 cm$^2$, given that $\Phi$ = 5 cm. After compressing the sample, the diameter, height, weight, and mass were measured in order to determine its density. The water content of each compressed sample was measured to determine the bulk density of the dry sample.

$$\rho d = \rho h / (1 + w) \tag{3}$$

where $w$ is the water content (%), so

$$C = \rho_d / \rho_{absolu} \tag{4}$$

Figure 7 shows the variation in dry density as a function of water content. The ideal water content for making this mixture was 13%.

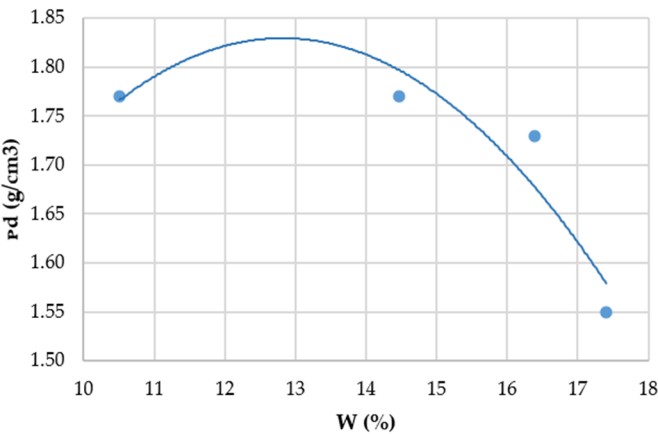

**Figure 7.** Variation of dry density as a function of water content.

*2.3. Mixture Design of CEB*

2.3.1. Sample Preparation

The production principle of CEB is to use compacted and mechanically shaped straw. According to the standard [51], various compression tests were carried out [66,67] in order to evaluate the compaction properties of the representative sediments.

The samples were consisted of the following (Table 3): 400 g of sediment (70% sediment from the port of Oran and 30% sediment from the port of Sidi Lakhdar), depending on the consistency curve as a function of percentage of each sediment (Figure 6), glass with 4 moles of sodium hydroxide solution (this solution was left to stand for 24 h. 30 g of glass powder was mixed in 100 mL of this solution and left in an agitator for 8 h, and the same solution was made again with 40 g of glass), 40 g of BFS, and 40 g of clay.

**Table 3.** The different raw brick formulations for the optimal water content of 13%.

| Formulation | Sediments (g) | Clay | | BFS | | Glass Powder Dissolved in NaOH Solution (%) | Water (g) |
|---|---|---|---|---|---|---|---|
| | | (%) | (g) | (%) | (g) | | |
| **Add 30 g of glass/100 mL of sodium hydroxide solution** | | | | | | | |
| 1 | | | | | | 2 | 25 |
| 2 | 400 | 10 | 40 | 10 | 40 | 4 | - |
| 3 | | | | | | 6 | - |
| 4 | | | | | | 8 | - |
| **Add 40 g of glass/100 mL of sodium hydroxide solution** | | | | | | | |
| 1 | | | | | | 2 | 32 |
| 2 | 400 | 10 | 40 | 10 | 40 | 4 | 12 |
| 3 | | | | | | 6 | - |
| 4 | | | | | | 8 | - |

The test consisted of mixing clay, sediment, blast furnace slag and a quantity of activated glass with a NaOH solution to make brick paste.

At all stages of this study, the mixtures were subjected to a compaction stress of 10 MPa. After remolding the test pieces, the diameter, height and mass of each sample were measured. After shaping, there was a drying phase in an oven at 60 °C. This was necessary in order to remove the residual water from the bricks produced. Subsequently, the molds produced were subjected to various mechanical tests.

2.3.2. Atterberg Limits

Figure 8 shows CASAGRANDE the diagram of plasticity proposed by the standard [51]. A strong swelling behavior characterized the sediments of the port of Oran,

which were considered low-plastic because they were easy to dry and showed good results in the manufacture of blocks.

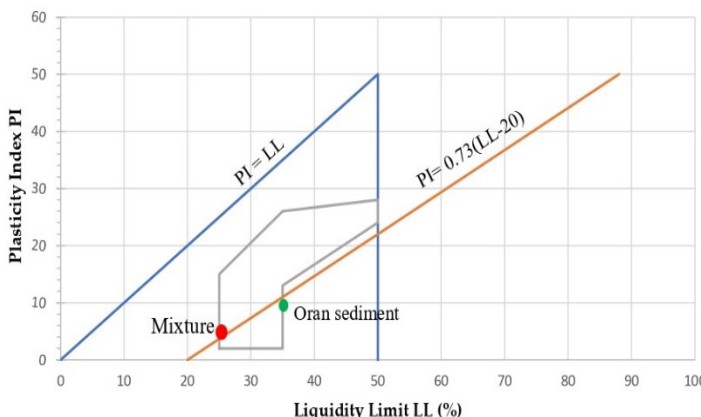

**Figure 8.** Atterberg limits of sediment.

2.3.3. Calculation of the Mass of the Mixture

The test consisted of mixing and homogenizing all the dry components, then a final amount of water was added to make the brick paste. In order to compact the samples, the mass of the mixture sufficient to fill a cylinder 5 cm in diameter and 10 cm in height, was calculated. Table 4 shows how this mass is determined.

**Table 4.** Determination of the mass of the compressed mixture.

| Cylindrical Specimens 5 × 10 | |
|---|---|
| **Processing:** | **10% BFS** |
| Unit volume (cm$^3$) | 98.2 |
| Number of test specimens to be made | 15 |
| Volume (cm$^3$) | 196.3 |
| $\rho_{d\,OPN}$ (kg/m$^3$) | 1795 |
| 98.5% $\rho_{d\,OPN}$ treated silt (kg/m$^3$) | 1768 |
| Dry mass of mixture to be sampled (kg) | 0.347 |
| to the water content $w_{OPN}$ (%) | 13 |
| Wet mass of mixture to be sampled (kg) | 0.382 |
| Dry mass of the mixture (kg) | 0.347 |
| Quantity of solution (g) | 45.108 |

The mass of the compressed mixture was 382 g.

## 3. Results and Discussion

### 3.1. Testing Samples Preparation

The behavior of the sediments under a static compaction test was used to study the impact of the compressive load on the properties of compacted earth. This process measures the amount of energy based on the properties of the soil and the amount of water contained. Compressive strength, capillarity, and water resistance were determined.

3.1.1. Compressive Strength

Samara et al. [68] considered compressive strength a good indicator of quality because it has a determining factor in the ability of a material to be used in construction. Simple compression tests were carried out on dry cylindrical test pieces by means of a mechanical press (INSTRON 30 kN). The speed of movement was 0.02 mm/s, corresponding to a pressure increase of 0.15 and 0.25 MPa/s, until the complete rupture of the test piece [51] (Figures 9 and 10). The values obtained were the average of 3 samples for each formulation.

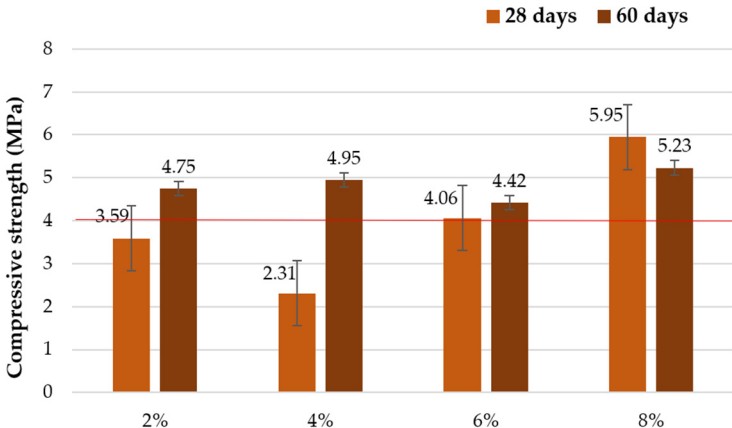

**Figure 9.** Measurement of compressive strength (MPa) by a adding 30 g of glass/100 mL of sodium hydroxide solution.

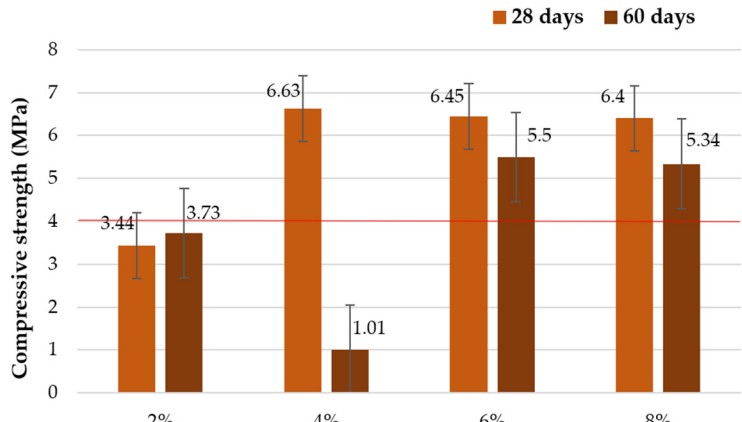

**Figure 10.** Measurement of compressive strength (MPa) by adding a 40 g of glass/100 mL of sodium hydroxide solution.

The results showed a variation of compressive strength by adding 30 g of glass/100 mL of sodium hydroxide solution between 2.31 MPa and 5.95 MPa, with the average being 4 MPa. These are very good values since it is known that the minimum compressive strength requirements for CEBs vary between 1.0 MPa and 2.8 MPa [69,70]. The maximum compressive strength obtained by adding 40 g of glass/100 mL of sodium hydroxide solution was 6.63 MPa. These values are of the same order as those found by Cottrell et al. [71]. According to Nshimiyimana et al., and Rivera et al. [67,72], the current CEB is classified in the CEB5 category (force of at least 5 MPa).

A concentration of 4 mol NaOH solution results in the formation of a homogeneous gel. This corresponds to an increase in compressive strength [73].

### 3.1.2. Water Absorption by Capillarity

- The water absorption coefficient Cb

In order to calculate the water absorption coefficient Cb (Table 5), the blocks were dried in an oven for 24 h until the mass was constant ($P0$). The blocks were then left to stabilize in the laboratory for 6 h. A smooth face was immersed until it was 10 mm underwater. After 10 min, the blocks were removed ($P1$) according to European standard XP P13-901 [51].

$$Cb = 100 M/S\sqrt{t} = 100\,(P1 - P0)/S\sqrt{t} \tag{5}$$

$M$: is the mass of water absorbed (g).
$S$: is the area of the submerged face (cm$^2$).
$t$: is the time of immersion block (min).

**Table 5.** Water absorption coefficient value.

| | Add 30 g Glass/100 mL Sodium Hydroxide Solution | | | |
|---|---|---|---|---|
| | **2%** | **4%** | **6%** | **8%** |
| **Cb** | 50 | 16.2 | 41.7 | 36.9 |
| | Add 40 g Glass/100 mL Sodium Hydroxide Solution | | | |
| | **2%** | **4%** | **6%** | **8%** |
| **Cb** | 59.3 | 28.7 | 58.7 | 41.1 |

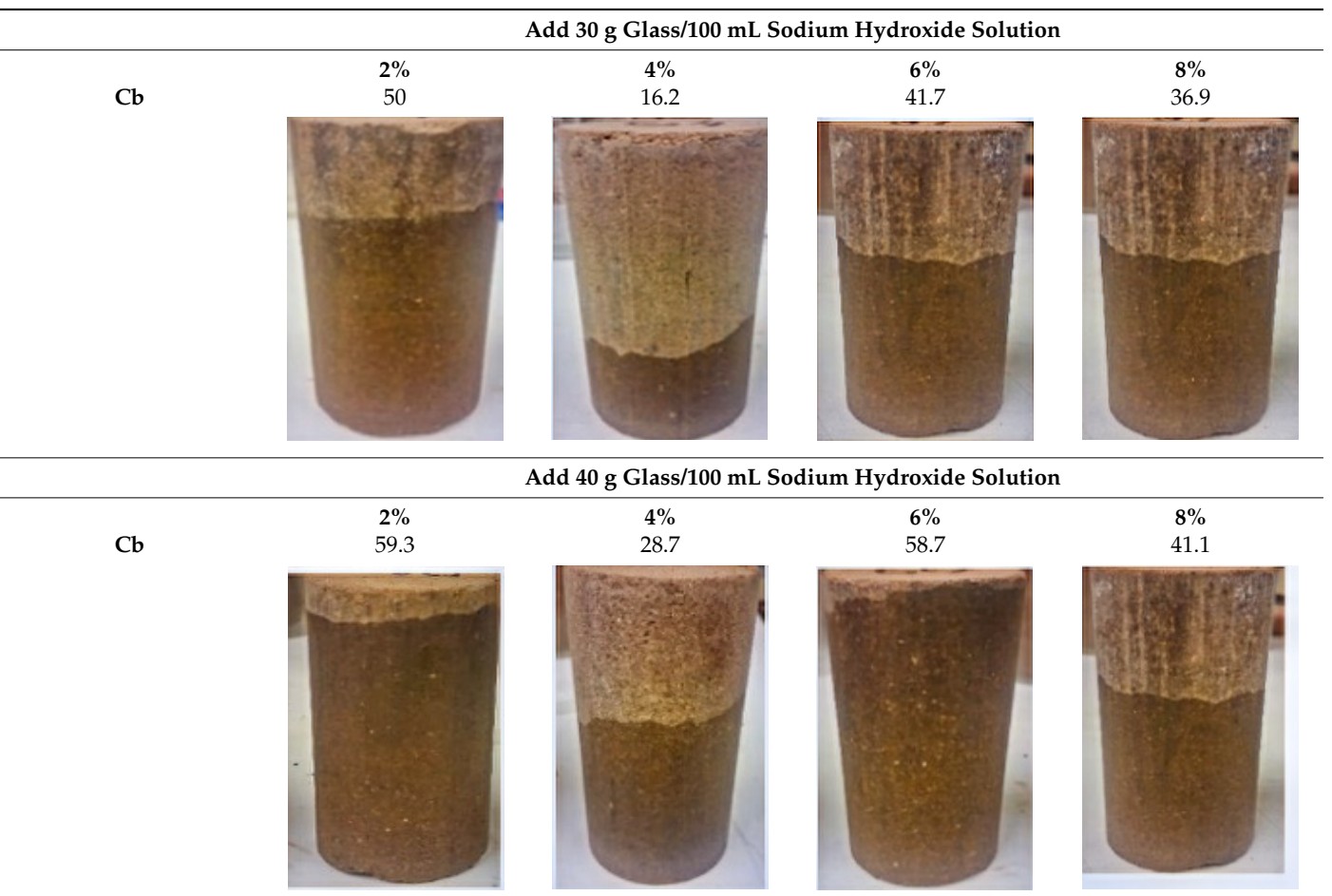

The absorption coefficient in the blocks which contained the 2, 6 and 8% solution was high due to the high porosity. The absorption coefficient increased as the bulk density decreased [71] but the blocks which contained 4% of glass with a soda solution were weakly capillary. These values were of same order as those found by Rivera et al. [72].

- Water absorption and packing density

The sample pore volume was measured by weighing it twice, first while dry, then again when all the voids were filled with water. The density of the water being known, the difference in mass gives the volume of water, which is equal to the pore volume if the entire porosity is filled with water. In order to ensure complete filling, the inhibition of the samples took place under vacuum: the pores were firstly emptied of air which in a vacuum enclosure, then degassed water was gradually introduced [73,74]. The porosity Nt of the sample is expressed by:

$$Nt = ((Wh - Ws)/Vt) \times 100 \qquad (6)$$

The measured water absorption values (Figure 11) obtained are the average of 3 samples for each formulation.

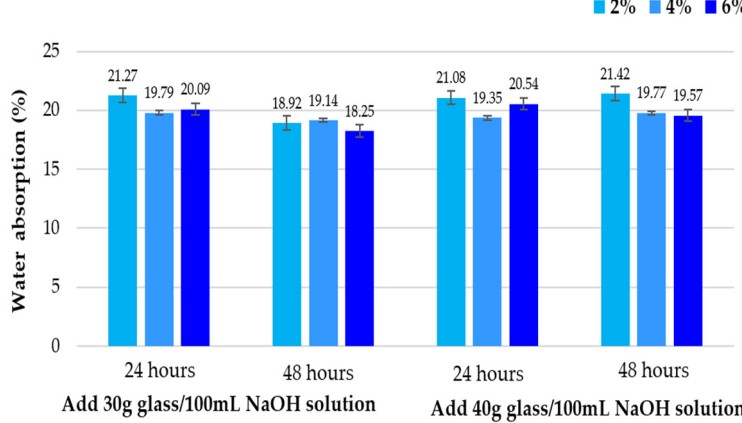

**Figure 11.** Water absorption value in 24 and 48 h.

The maximum recorded rate of porosity was when 2% of solution was added. On the other hand, the addition of 6% of solution lowered the porosity to 18.25% in 48 h, which means that the quantity of solution and the size of the pores were the most important factors in the measurement of porosity. Reduced porosity gives a higher compressive strength and therefore a better quality. These values are favorable when compared to clay bricks (0–30%), concrete blocks (4–25%), and calcium silicate bricks (6–16%) [69].

Table 6 shows the compaction densities of the blocks. Blocks containing 2% of 30 g/100 mL of solution have a less density at 2.62 g/cm$^3$, while blocks containing 8% of 30 g/100 mL of solution and 6% of 40 g/100 mL of solution have the highest density at 2.75 g/cm$^3$.

**Table 6.** Obtained density values.

| **Addition of 30 g/100 mL of the Solution** | **2%** | **4%** | **6%** | **8%** |
|:---:|:---:|:---:|:---:|:---:|
| Density (g/cm$^3$) | 2.62 | 2.69 | 2.68 | 2.75 |
| **Addition of 40 g/100 mL of the Solution** | **2%** | **4%** | **6%** | **8%** |
| Density (g/cm$^3$) | 2.69 | 2.72 | 2.75 | 2.67 |

### 3.1.3. Water Resistance

For the measurement of water resistance, the blocks are completely immersed in water for 24 h (Table 7).

Blocks that contain 8% glass solution break in water, which means that, to create more resistant blocks, the amount of glass solution added should be small.

### 3.1.4. Scanning Electron Microscopy (SEM)

The SEM images taken by a JEOL device with an acceleration voltage of 5 KV and, for different scales of 40, 100 and 200, show that the CEB which contains more glass solution, no longer resists because it is porous. On the other hand, the CEB containing 4% of the glass solution did not contain many pores, which increased means in compressive strength (Figures 12 and 13).

According to microscopic observations and chemical composition, the reason why the reaction product was tightly bound to the surface of the CEB particles produced by the NaOH -activated glass powder was the presence of a large proportion of Silicium (Si) and calcium (Ca) and a low proportion of sodium (Na). The percentage of sodium increases the percentage of glass increases because it contains 13.58% of sodium oxide, with 8% added to the NaOH solution. Sun et al. and Maraghechi et al. [75,76] conclude that the products with a higher calcium content are denser, solid, and more stable by volume.

**Table 7.** Water resistance test in blocks after 24 h.

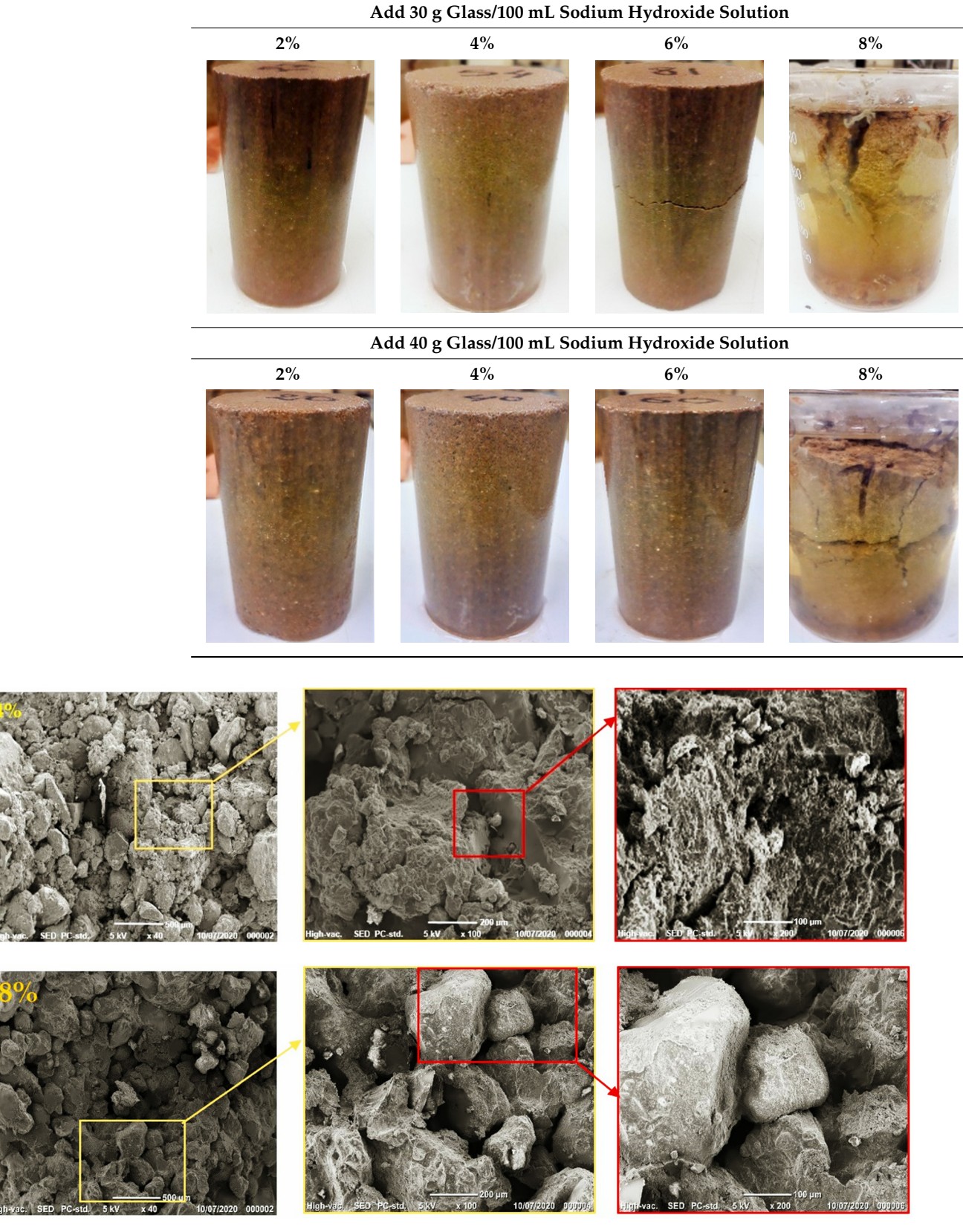

**Figure 12.** Scanning electron microscope (SEM) images of the mixture with 30 g/100 mL NaOH solution added.

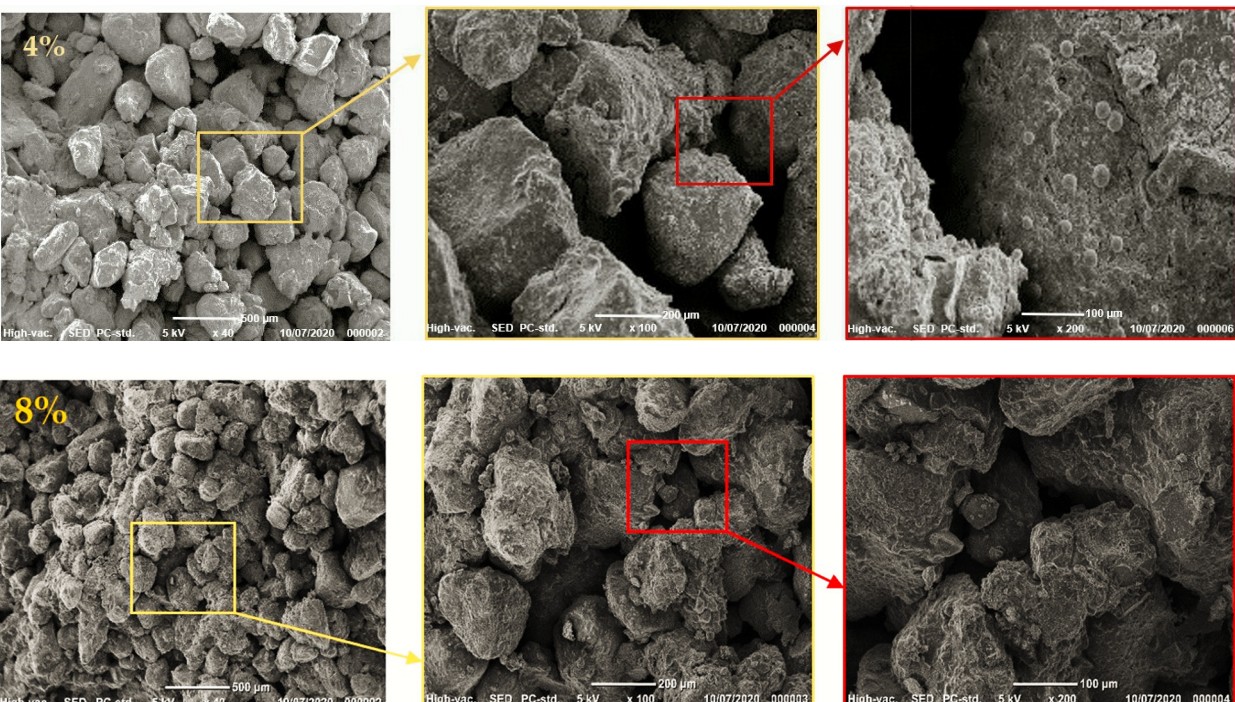

**Figure 13.** Scanning electron microscope (SEM) images of the mixture with 40 g/100 mL NaOH solution added.

## 4. Conclusions

Soil stabilization is extremely important in the manufacture of the CEB to ensure good mechanical properties. Glass waste is an excellent choice for soil treatment. The advantage of this stabilizer is linked to the small amount being added and its ecological use.

Taking this into account, the increase to the compressive strength may be a function of the amount of glass solution added. The mechanical behavior of the blocks depends on the grain size, nature of the sediments, water content, binders (slag and glass solution), and the clay. The compressive strength of the blocks also depends on their density, porosity, and the size distribution of the pores.

According to the information gathered, the compressive strength after 60 days was higher than that after 28 days. For the various mixtures prepared, it was found that an increase to the content of the glass solution resulted in an increase to the compressive strength values. Lower empty volume is linked to better strength and therefore much better quality.

The difference in compressive strength clearly and varied with the various glass solution ratios. Indeed, for maximum compressive strength, the addition of 4% glass solution produces the best mixture, while a lower percentage of glass powder activated with NaOH solution provides greater compressive strength with less porosity.

**Author Contributions:** Conceptualization, S.L. and W.M.; methodology, S.L., W.M. and N.-E.A.; validation, S.L., W.M. and A.K.; formal analysis, W.M. and S.L.; investigation, S.L. and W.M.; resources, N.-E.A.; data curation, S.L. and W.M.; writing—original draft preparation, S.L.; writing—review and editing, S.L., W.M. and A.K.; visualization, S.L., W.M. and A.K.; supervision, W.M., A.K. and N.-E.A.; project administration, W.M., N.-E.A. and A.K.; funding acquisition, W.M. All authors have read and agreed to the published version of the manuscript.

**Funding:** This research received no external funding.

**Institutional Review Board Statement:** Not applicable.

**Informed Consent Statement:** Not applicable.

**Data Availability Statement:** The data presented in this study are available on request from the corresponding authors.

**Acknowledgments:** This study was supported by the Ministry of Higher Education and Scientific Research of Algeria. The authors thank Damien BETRANCOURT for the physical analyses, and IMT Nord Europe.

**Conflicts of Interest:** The authors declare no conflict of interest.

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
