# Peer review of "Formulation of Compressed Earth Blocks Stabilized by Glass Waste Activated with NaOH Solution"

_sustainability, doi:10.3390/su14010102_

Round 1

Reviewer 1 Report

The manuscript shows a very interesting and novel topic. The English have to be reviewed by an expert native person. The manuscript must write in the third person, not use pronouns like I, and we. For all figures, improve the resolution. Although the article shows interesting results, the document is disorganized, and the analysis by the authors is poor. I recommend that you review it by a person who is an expert in writing scientific articles and resubmit it to the journal after the corrections.

Specific comments:

For all figures, improve the resolution.

In all documents please using a super index for the m3 unit

Abstract: correct the NaOH concentration, resistance is not correct, please use compressive strength.

Introduction:

Line 42, 45, 46, 51- please correct the citation form. Thiago et al (2017), Narayanaswamy et al. (2020) is not correct.

Materials and Methods:

Line 69,71,75- please correct super index in m3 units.

Table 1- please use IS units for specific surfaces, please explain what is VBS, ES?

Table1: what about XRF, the particle size distribution for Blast furnace slag and glass waste?

Unify tables 1,2, and 3 into one.

Line 94- please using numbers: 24 h

Line 98- please indicate the EU number? used following of [44,63]

Line 100- rewriting: is not necessary the phrase "to work out its mass loss"

Line 102- what is OM? maybe metal oxide

Line 119-120: This is not true, what is your reference ?. rectify. The typical SiO2 / Al2O3 mass ratio for kaolin and montmorillonite is not 9.5-10.0. The real SiO2 / Al2O3 mass ratio is between 1.5-2.5.

Line 149: the correct word is TGA, not GTA, please rectify

Line 150: for TGA the correct word is -mass loss-.

The is a lack of references based on literature for explaining TGA-DTG results.

DTG- What about endothermic peaks around 300 ° C and 500 ° C.?

Line 159- what is CEB?

you define this in line 204. You can define CEB in introductions or abstracts.

Line 210- 2.3 mix design: Based on what criteria did you choose to mix 70% Oran sediment and 30% Sidi sediment?

line 210- you talk about concentration in molarity for NaOH solution, but in Table 5 you show% wt. Please clarify

Figures 2 and 8 are equal.

Line 255- is not resistance, the correct term is -Mechanical Strength-. Resistance is used for example acid resistance, water resistance, in durability. For mechanical properties, the correct word is strength.

Figure 10- UCS 28d and UCS 60days: can you unify for a better understanding

Figure 11. idem

Line 265-line 268: expand the discussion on the effect of the alkalinity of the NaOH solution on mechanical strength.

How many samples did you average to determine mechanical strength and porosity? Place standard deviation in graphs 10, 11, and 12.?

Line 282- you talk about porosity but the results are later. You can move it to this section.

Line 313- it is wrong titled, your images are about SEM (scanning electronic microscopy), rectify!

Table 10- rectify, are SEM images not optical images.

Reviewer 2 Report

Your article contains lot of experimental values and methods and is a good example of modern use of alternative materials for a sustainable society.

Reviewer 3 Report

The article contains interesting information on soil stabilization. However, there is no more widely conducted research on stabilized propb. After they are completed, the article will gain in an ankish value.

Round 2

Reviewer 1 Report

The research is very interesting, but the way in which the results are presented and discussed can be improved for greater public understanding. Specific comments sent below:

1)The English have to be reviewed by an expert native person.

2)Abstract-line 18:  define CEB (compressed earth block)

3)Materials and Methods:

Li 88- Why the water content for Glass powder and BFS are 39% and 33%, respectively. Is those sludge?

4)Note that the comma (,) is not used to separate decimal places.

5)line 71- Rectify, rate of 69.704 m3/ day

6) line 73- Rectify, 120.000 m3 excavation material

7) line 77- Rectify, 280.000 m3

8)XRD results: line 141- for quartz and lime minerals identified by XRD, please indicate de # PDF or # ICSD file?. Are you sure is lime or calcite ?.

9)XRD results: why you don’t identify clay minerals on XRD (Fig. 3) as kaolin, illite, ?. nevertheless, on TGA results (line 184-185) you say that the peak around 500ªC on the DTA curve is by kaolinite and illite present. ? and 2.3.2 section, the sediment of Oran is considered plastic?

10)for TGA:  the correct word is -mass loss-.

11)Table 3- replace comma (,) with a dot (.)

 12)Line 295- Line 298, clarify this explanation if for CEB based on fly ash with high carbon content.

13)SEM results: please increase the size of the images. I think the images are incomplete, please correct them. Expand the discussion of SEM results, relying on the literature already consulted.
